# Home settings are associated with less functional decline among older adults compared to community-care foster homes and skilled nursing facilities in Hawaii

Hua Zan[1]*, Yanyan Wu[2], Yan Luo[3], John P. Barile[4], Joshua R. Holmes[5], Joy Agner[6]

1 Center on the Family, University of Hawaii at Manoa, Honolulu, Hawaii, United States of America, 2 Office of Public Health Studies, University of Hawaii at Manoa, Honolulu, Hawaii, United States of America, 3 Social Science Research Institute, University of Hawaii at Manoa, Honolulu, Hawaii United States of America, 4 Department of Psychology, University of Hawaii at Manoa, Honolulu, Hawaii, United States of America, 5 Med-QUEST Division, Hawaii Department of Human Services, Honolulu, Hawaii, United States of America, 6 Chan Division of Occupational Science and Occupational Therapy, University of Southern California, Los Angeles, California, United States of America

* hzan@hawaii.edu

## Abstract

Medicaid-funded home and community-based services (HCBS) allow older adults with disabilities to avoid long-term institutionalization in nursing homes or hospitals. Past research has shown mixed results on the positive impacts of HCBS. These inconsistent results may stem from studies combining varied HCBS settings, obscuring their differential impacts on older adults' health and well-being. In Hawaii, HCBS settings primarily include private residences and community care foster homes. There is very little research on adult foster homes, and it remains unclear whether adult foster homes are associated with differential rates of functional decline over time compared to private homes or nursing homes. This research contributes to these literature gaps by comparing functional decline (measured using Hawaii Medicaid level-of-care assessments) across three settings: private homes, adult foster homes, and nursing homes from 2014 to 2021. Among 5,315 dual eligible Medicaid recipients, we found distinct characteristics in initial placement. Individuals placed at home were younger and had lower functional impairment scores compared to individuals in foster homes or nursing homes. To increase comparability despite these differences, we matched older adults (n = 852) on baseline functional status, age, sex, marital status, and race/ethnicity using propensity score matching and performed sensitivity analyses on cognitive status. After matching, linear mixed-effects modeling revealed a notably slower rate of functional decline at home compared to nursing homes or foster homes. Individuals at home had fairly stable functional status (low deterioration) over the eight years. Nursing home residents had the fastest rate of decline, followed closely by individuals in foster homes. These findings of the varying functional

**Data availability statement:** This study used Hawaii's level of care assessment data and Medicaid administrative data. The data was made available to the researchers by the Med-QUEST Division, Hawaii Department of Human Services based on our role as evaluators for the 1115 Waiver Demonstration. Data access was granted through a Business Associate Agreement between the University of Hawaii and the Hawaii Department of Human Services. Based on this agreement, the researchers are not authorized to reshare the data used in this study. Data requests may be sent to the State Health Planning and Development Agency in Hawaii (SHPDA@doh.hawaii.gov).

**Funding:** This work was supported by State of Hawaii Department of Human Services, United States of America. The funder provided all data for analysis, offered context on program implementation, reviewed analyses and the final manuscript, and supported the decision to publish.

**Competing interests:** The authors have declared that no competing interests exist.

outcomes across care settings can inform policymakers, families, and caregivers in selecting effective care options.

## Introduction

Many older adults, including those with disabilities or functional limitations, prefer to age in place for as long as possible [1,2]. Home- and community-based services (HCBS) are designed to allow individuals with disabilities to receive care in community and residential settings, sustain their functioning, and continue living independently [3]. Given that functional decline poses a significant threat to independent living for older adults [4], understanding the extent to which HCBS can slow functional decline is critical. As functional impairment increases, caregiver burden increases [5], long-term care costs increase [6], and older adults participate less in valued social roles in home and community [7]. Further, as functional impairment increases, older adults are at risk of institutionalization in skilled nursing facilities and hospitals [8], which can separate them from important social supports and reduce overall quality of life.

Theoretically, factors that affect functional status include personal characteristics and contextual or environmental factors. Contexts that older adults live in are particularly important for their health and well-being, as contextual factors play a greater role as opposed to personal characteristics as people age and their competence declines [9,10]. As people age, most of them spend more of daily life at home or in the community [11,12] and prefer staying in home and community-based service (HCBS) settings [1]. HCBS settings tend to provide older adults a familiar environment and preserve their autonomy and social connection [13]. However, compared with nursing homes, HCBS settings might not always be adequately equipped to meet the medical and safety needs of individuals [14], especially for those with decreased functional capacities [13]. As such, the extent to which HCBS settings are associated with better or worse functional status among older adults is an empirical question.

The literature lacks compelling evidence to establish the superiority of HCBS settings over nursing homes concerning patient outcomes [15]. Systematic reviews yielded mixed results comparing the effects of HCBS versus institutional care on health outcomes [16]. While one study favored home care with support [17], others found HCBS settings were not superior to institutional settings in terms of functional status, healthcare use, and mental health [18,19]. The failure to consider within-group differences in HCBS population and selection bias may contribute to the inconclusiveness of the literature. Older adults receiving long-term services and supports are a heterogeneous population, and individuals with certain characteristics often gravitate toward particular settings. Since most prefer aging in place, individuals living at HCBS settings tend to have lower mortality risk, less comorbidities and less complex needs, higher functional status, and use less intensive health care services compared to nursing home residents [20–23]. Consequently, treating the HCBS population as a homogeneous group and failing to consider selection bias may mask the effects of HCBS on health. Furthermore, family and patient decisions about care

settings can be affected by availability of services, providers' recommendations, and housing and care resources within their area [24–27]. Therefore, many selection factors affect where placement occurs. A review of studies comparing the health effects across settings critiqued most of these studies are susceptible to bias, particularly attributable to factors such as the absence of randomization and baseline imbalances [16]. Furthermore, studies often lump different HCBS settings together [14], and do not examine differential impacts of various home and community based options, such as adult foster homes, assisted living facilities, or private residences.

This study attempts to fill the gap by analyzing longitudinal data, addressing baseline imbalances, and comparing the functional status of dually eligible Medicaid recipients 65 years or older in three settings: home, foster home, and nursing home in Hawaii. To qualify for Medicaid in Hawaii, applicants must be state residents, meet specific citizenship or immigration requirements, and demonstrate limited financial resources. Income and asset limitations vary depending on the specific Medicaid program. In addition, accessing long-term care services and supports, such as HCBS and nursing home care, requires an assessment to confirm the functional need for such care. Upon approval following this evaluation, which is primarily based on activities of daily living scores [28], individuals gain access to tailored services and supports [29].

Our sample consists of adults over 65 who are dually eligible for Medicare and Medicaid and qualify for HCBS due to disability. This population experience multiple, intersecting risk factors. Research indicates that dual eligible individuals tend to experience physical and cognitive impairment, high prevalence of chronic conditions, extensive social welfare needs, and high utilization of health care services and costs [14,30]. Furthermore, racial/ethnic minority groups are disproportionately represented within this population [14]. Additionally, among dual eligibles who all face economic insecurity, our sample has highlighted disability—indicated by high scores on functional impairment testing—as a significant factor that exacerbates existing social and medical risk factors [31]. Given that addressing the unique needs of individuals with multiple, complex social and physical health needs is the primary purpose of HCBS, it is crucial to consider and empirically examine possible factors within HCBS that might influence health outcomes among older adults.

Home and foster homes are the two major HCBS settings in Hawaii, and nursing homes are used as a comparison group. Unlike institutional settings, foster homes are meant to provide residents a family-oriented environment. In Hawaii, these homes can accommodate up to three residents with at least one Medicaid recipient per foster home [32]. In 2023, there were more than 1,000 licensed community care foster homes in Hawaii [33], yet the research on adult foster homes is very thin both within and outside of Hawaii. Adult foster homes, sometimes called board and care homes, had been designated HCBS settings in at least 14 states by 2009 [34]. This report found that adult foster homes are declining in some states, and are often underfunded as they receive low reimbursement rates from Medicaid, at times too low to cover costs of care [34]. The majority of research on resident outcomes within adult foster homes has been conducted in Veteran's Administration (VA) studies, and VA foster homes are likely distinct from most Medicaid funded adult foster home in the payment model, oversight, caregiver training and therefore likely the quality of care [35–37]. Thus, this study is of particular relevance and significance for our understanding of the degree to which non-institutional settings, such as foster homes and private residences, contribute to improved health and well-being in the context of HCBS expansion in the past decades with a shift in funding priorities from institutional care to home and community-based care [14]. To compare across settings, our study did not include assisted living residences because very few individuals receiving HCBS in Hawaii are placed in assisted living facilities.

Because our focus is on the impact of contextual factors, namely residential settings, this study is guided by the conceptual framework for studying Context Dynamics in Aging (CODA) developed by Wahl and Gerstorf [10], which provides a comprehensive framework for studying the role that contexts play in affecting the health and well-being of older adults. The tenet of this framework suggests setting plays an important role in supporting or constraining the functioning of older adults. As such, we hypothesize that HCBS settings (i.e., home or foster home) are associated with different patterns of functional status among older adults compared to nursing homes. We also expect to identify heterogeneity within HCBS settings and hypothesize the patterns of functional status among individuals receiving care at home are different from

those residing in foster homes. However, since setting, as the context, is associated with different opportunities and constraints for successful aging [10], the extent to which settings are associated with better or worse functional status among older adults is an empirical question.

## Materials and methods

### Data and sample

We analyzed Hawaii's level of care (HILOC) assessment data and Medicaid administrative data from 2014 to 2021. HILOC data are collected from a level of care (LOC) assessment form (Form 1147), which is used to assess an individual's functional status and LOC needs. In addition to the ability in activities of daily living (ADL)—commonly referred as functional status in the literature [38,39], the LOC assessment also covers cognition, communication, and behaviors [40]. This tool has not been formally assessed to determine its effectiveness in evaluating the functional status of individuals. The LOC assessment is performed by a physician, registered nurse, or primary care provider within a managed care organization or via delegated authority. The cumulative points constitute the individual's total LOC score with higher points indicating higher severity in functional limitations. After the initial assessment, reassessments should happen every 12 months, when individuals experience significant changes in health or circumstance, or when they request reassessments. Because HILOC data includes limited demographic information, we added marital status and race/ethnicity from the administrative data. We last accessed the data on August 27, 2024, and we do not have access to information that could identify individual participants during or after data collection. This study was reviewed and determined to be not human subjects research by University of Hawaii Office of Research Compliance.

Given our interest in older adults, we restricted the sample to Medicaid recipients aged 65 or older, thus individuals who are dually eligible for Medicaid and Medicare, as well as eligible for long-term services and supports based on functional status. To track functional status over time, we excluded those without follow-up assessments or with follow-up for less than two years. To not obfuscate the impact of setting on functional status over time, we also excluded members who had changed settings within this period. We note that excluded members showed different characteristics compared to those staying in the same settings (Table 1). The final sample includes 5,315 dually eligible Medicaid recipients (sample selection process in S1 Fig). The average years of follow-up was 3.6 years with a standard deviation of 1.4 years, and the average number of repeated assessments was 4.7 with a standard deviation of 1.6 assessments.

### Variables

To measure functional status, we constructed the total LOC score by summing up the points assigned to assess an individual's physical, mental, and cognitive abilities. The minimum and maximum total LOC scores are zero and 38 respectively. The independent variable is the setting (i.e., home, foster home, and nursing home). The covariates include age (years), sex (male and female), marital status (married/partnered, divorced/separated, single, widowed, and unknown), and race/ethnicity (White, Native Hawaiian including part Native Hawaiian, other Pacific Islander, Filipino, Japanese, Chinese, and other) reported at baseline.

### Analytical approach

The primary goal of this study is to assess the extent to which the LOC scores of individuals living at home, foster homes, versus nursing homes had the same rate of change over time if their average LOC scores at baseline were the same. To ensure the samples were balanced across settings, we used a propensity score method to match their baseline characteristics including LOC score and covariates. Because the matching method can only match two groups at a time, we first matched individuals at home to those living in foster homes and nursing homes respectively. Then, the two separate matched samples were linked by the individuals in the home setting, ensuring that all three settings were matched. We used a 1:1 ratio and a small caliper value of 0.01 to ensure the matching quality. The matched sample included 852

**Table 1. Baseline characteristics of the unmatched and matched samples of individuals staying in the same setting with ≥ 2-year of follow-up and excluded sample.**

| Variable | | N | Unmatched sample of individuals staying in the same setting with ≥ 2-year follow-up (n = 5,315) | | | Excluded sample* (n = 3,094) | P | Matched sample of individuals staying in the same setting with ≥ 2-year follow-up (n = 852) | | | | P |
|---|---|---|---|---|---|---|---|---|---|---|---|---|
| | | | Home | Foster home | Nursing home | | | n | Home | Foster home | Nursing home | |
| All | | | 2,354 (28.0%) | 1,185 (14.1%) | 1,776 (21.1%) | 3,094 (36.8%) | | | 284 (33.3%) | 284 (33.3%) | 284 (33.3%) | |
| Sex | Male | 1,512 (28.4%) | 650 (27.6%) | 380 (32.1%) | 482 (27.1%) | 1,055 (34.1%) | <0.0001 | 247 (29.0%) | 84 (29.6%) | 85 (29.9%) | 78 (27.5%) | 0.783 |
| | Female | 3,803 (71.6%) | 1,704 (72.4%) | 805 (67.9%) | 1,294 (72.9%) | 2,038 (65.9%) | | 605 (71.0%) | 200 (70.4%) | 199 (70.1%) | 206 (72.5%) | |
| Age at baseline | Mean±SD | 80.7±9.7 | 77.4±8.8 | 82.1±9.8 | 84.1±9.4 | 80.4±9.2 | <0.0001 | 81.9±9.8 | 81.9±9.9 | 80.8±9.8 | 83.1±9.7 | 0.018 |
| Marital status | Married/partnered | 630 (11.9%) | 490 (20.8%) | 30 (2.5%) | 110 (6.2%) | 339 (11.0%) | <0.0001 | 52 (6.1%) | 23 (8.1%) | 16 (5.6%) | 13 (4.6%) | 0.197 |
| | Divorced/separated | 1,254 (23.6%) | 686 (29.1%) | 252 (21.3%) | 316 (17.8%) | 658 (21.3%) | | 188 (22.1%) | 58 (20.4%) | 72 (25.4%) | 58 (20.4%) | |
| | Single | 714 (13.4%) | 276 (11.7%) | 252 (21.3%) | 186 (10.5%) | 397 (12.8%) | | 124 (14.6%) | 44 (15.5%) | 46 (16.2%) | 34 (12.0%) | |
| | Widowed | 1,533 (28.8%) | 658 (28.0%) | 305 (25.7%) | 570 (32.1%) | 711 (23.0%) | | 272 (31.9%) | 91 (32.0%) | 87 (30.6%) | 94 (33.1%) | |
| | Unknown | 1,184 (22.3%) | 244 (10.4%) | 346 (29.2%) | 594 (33.4%) | 989 (32.0%) | | 216 (25.4%) | 68 (23.9%) | 63 (22.2%) | 85 (29.9%) | |
| Race/ethnicity | White | 1,173 (22.1%) | 650 (27.6%) | 218 (18.4%) | 305 (17.2%) | 672 (21.7%) | <0.0001 | 186 (21.8%) | 73 (25.7%) | 59 (20.8%) | 54 (19.0%) | 0.014 |
| | Native Hawaiian | 410 (7.7%) | 205 (8.7%) | 70 (5.9%) | 135 (7.6%) | 277 (9.0%) | | 71 (8.3%) | 23 (8.1%) | 24 (8.5%) | 24 (8.5%) | |
| | Other Pacific Islander | 272 (5.1%) | 225 (9.6%) | 12 (1.0%) | 35 (2.0%) | 128 (4.1%) | | 15 (1.8%) | n/r (n/r) | n/r (n/r) | n/r (n/r) | |
| | Filipino | 842 (15.8%) | 454 (19.3%) | 184 (15.5%) | 204 (11.5%) | 516 (16.7%) | | 149 (17.5%) | 50 (17.6%) | 64 (22.5%) | 35 (12.3%) | |
| | Japanese | 1,027 (19.3%) | 114 (4.8%) | 389 (32.8%) | 524 (29.5%) | 416 (13.4%) | | 163 (19.1%) | 41 (14.4%) | 60 (21.1%) | 62 (21.8%) | |
| | Chinese | 394 (7.4%) | 236 (10.0%) | 68 (5.7%) | 90 (5.1%) | 260 (8.4%) | | 58 (6.8%) | 22 (7.7%) | 15 (5.3%) | 21 (7.4%) | |
| | Other | 1,197 (22.5%) | 470 (20.0%) | 244 (20.6%) | 483 (27.2%) | 825 (26.7%) | | 210 (24.6%) | 69 (24.3%) | 56 (19.7%) | 85 (29.9%) | |
| LOC score | Mean±SD | 16.2±8.0 | 9.7±6.8 | 21.9±3.4 | 21.0±4.5 | 15.9±7.1 | <0.0001 | 20.5±3.8 | 20.9±3.9 | 20.5±3.4 | 20.0±4.1 | 0.012 |

Notes:

*Excluded sample includes individuals who had changed settings from 2014 to 2021 and those without follow-up assessments or with follow-up for less than two years. n/r refers to no reporting of counts which are less than 11 for any Medicaid data.

individuals with 284 in each setting. We first summarized baseline characteristics for the unmatched sample (n = 5,315) and the matched sample (n = 852) by setting. We then conducted Chi-square and one-way ANOVA tests to examine differences by setting. Lastly, we used linear mixed-effects (LME) models with random intercept to assess the longitudinal change of LOC score by setting with the interaction effects between setting and years of follow-up controlling for covariates at baseline for both unmatched and matched samples. The LME model is specified as follows:

$$LOC_{ij} = \beta_0 + \beta_1 \times year_{ij} + \beta_2 \times setting_i + \beta_3 \times (year_{ij} * setting_i) + \beta \times covariates + b_i + \varepsilon_{ij},$$

where $i = 1, 2, \ldots, n$ denotes the $i$th individual, $j = 1, 2, \ldots, n_i$ denotes $j$th repeated measures for the $i$th individual with $n_i$ repeated measures, $b_i$ denotes random intercept for the $i$th individual, and $\varepsilon_{ij}$ is the within individual random errors. The term $\beta \times \textit{covariates}$ denotes a vector of slopes for all covariates controlled in the model. The fixed-effect intercepts in LME models ($\beta_0$) estimate the average baseline LOC scores and the fixed-effect slopes ($\beta_1$, $\beta_2$, and $\beta_3$) assess the longitudinal change in LOC score per year for three settings. The assessment form covers various aspects of LOC needs such as communication, memory, mobility, and dressing and grooming. To determine how these needs vary across settings, we compared the average scores for each assessment item by setting after matching. Given the established link between cognitive decline and increased care needs and worsened health outcomes [41,42] we further aggregated cognition items (Items V-VI in Form 1147) and ADL items (Items VII-XIII in Form 1147). By comparing the scores of these two broad categories of care needs across settings, we aim to determine the influence of the change in cognition and functional status on shifts in LOC scores within the three settings.

## Results

Table 1 shows the baseline characteristics across settings in the unmatched and matched samples. In the unmatched sample, among members who stayed in the same setting with at least two years of follow-up, individuals residing at home tend to be younger, married, have higher percentages of whites, other Pacific Islanders, Filipinos and Chinese members, have a lower percentage of Japanese members, and have lower LOC scores, compared to those living in foster homes or nursing homes.

After matching, the baseline characteristics demonstrate a greater balance across settings when compared to the unmatched sample. There were no statistically significant differences in sex and marital status across settings after matching. For race/ethnicity and LOC score, despite the statistical significance, the differences across settings in the matched sample are much smaller compared to the unmatched sample. Note that, compared to individuals residing at home in the unmatched sample, those residing at home in the matched sample tend to be older (81.9 vs. 77.4), have a lower percentage of being married (8.1% vs. 20.8%), and a higher average LOC score (20.9 vs. 9.7).

Table 2 presents the results in the matched and unmatched sample of LME models with the intercept indicating the average LOC score at baseline and the slope indicating the change in LOC score per year. Before matching, residents in the three settings had different LOC scores at baseline. The gaps narrowed after matching. In the matched sample, nursing home residents had slightly lower LOC points compared to nursing home residents and residents at home at baseline. There was no difference between residents at home and those living in foster homes.

In terms of changes in LOC scores, the difference in slopes indicates that LOC scores increased over time in the unmatched sample. The rate of increase (e.g., rate of functional decline) was the largest among nursing home residents, followed by those living at home and then in foster homes. In the matched sample, however, the LOC score was stable over time for those residing at home, whereas it increased for foster home and nursing home residents. The rate of increase was higher for nursing home residents compared to foster home residents. The above findings are also illustrated in Fig 1.

The descriptive analyses on the components of the LOC assessment showed that members score each item differently by setting at baseline after matching (Table 3). For example, members living in foster homes had higher scores for vision/hearing/speech, cognition, and bladder function/continence on average compared to individuals residing at home or in nursing homes. Members in nursing homes had higher scores for transferring and mobility compared to those in the other two settings. Member residing at home had higher scores in communication, bathing, dressing and personal grooming. When items were aggregated, members in foster homes had higher scores for cognition compared to those at home or in nursing homes. However, no significant difference was found in the average scores for ADLs by setting. Additionally, regardless of settings, the average scores for cognition are much lower compared to the maximum allowable points because of the low scores for "Mental Status/Behavior" (Item VI).

**Table 2. Differences in baseline LOC scores and changes in LOC scores by setting in the linear mixed-effect modeling results using the unmatched and matched sample.**

| | Unmatched Sample | | Matched Sample | |
|---|---|---|---|---|
| **Intercepts and differences in intercepts** | | | | |
| **Setting** | **Estimate (95% CI)** | **P** | **Estimate (95% CI)** | **P** |
| Home | 9.8(9.6,10.0) | <0.0001 | 20.7(20.2,21.1) | <0.0001 |
| Foster home | 22.1(21.8,22.4) | <0.0001 | 20.9(20.5,21.4) | <0.0001 |
| Nursing home | 20.8(20.5,21.0) | <0.0001 | 20.2(19.7,20.6) | <0.0001 |
| **Comparison between settings** | | | | |
| Nursing home vs. Home | 11.0(10.6,11.3) | <0.0001 | −0.54(−1.17,0.10) | 0.0993 |
| Nursing home vs. Foster home | −1.3(−1.7,-0.9) | <0.0001 | −0.79(−1.42,-0.15) | 0.0156 |
| Foster home vs. Home | 12.3(11.9,12.7) | <0.0001 | 0.25(−0.39,0.89) | 0.4387 |
| **Slopes and differences in slopes** | | | | |
| **Setting** | **Estimate (95% CI)** | **P** | **Estimate (95% CI)** | **P** |
| Home | 0.45(0.42,0.48) | <0.0001 | 0.05(−0.03,0.13) | 0.2621 |
| Foster home | 0.40(0.37,0.44) | <0.0001 | 0.48(0.42,0.55) | <0.0001 |
| Nursing home | 0.65(0.61,0.68) | <0.0001 | 0.72(0.65,0.79) | <0.0001 |
| **Comparison between settings** | | | | |
| Nursing home vs. Home | 0.20(0.15,0.24) | <0.0001 | 0.67(0.57,0.78) | <0.0001 |
| Nursing home vs. Foster home | 0.24(0.19,0.29) | <0.0001 | 0.24(0.14,0.33) | <0.0001 |
| Foster home vs. Home | −0.04(−0.09,0.01) | 0.0886 | 0.44(0.33,0.54) | <0.0001 |

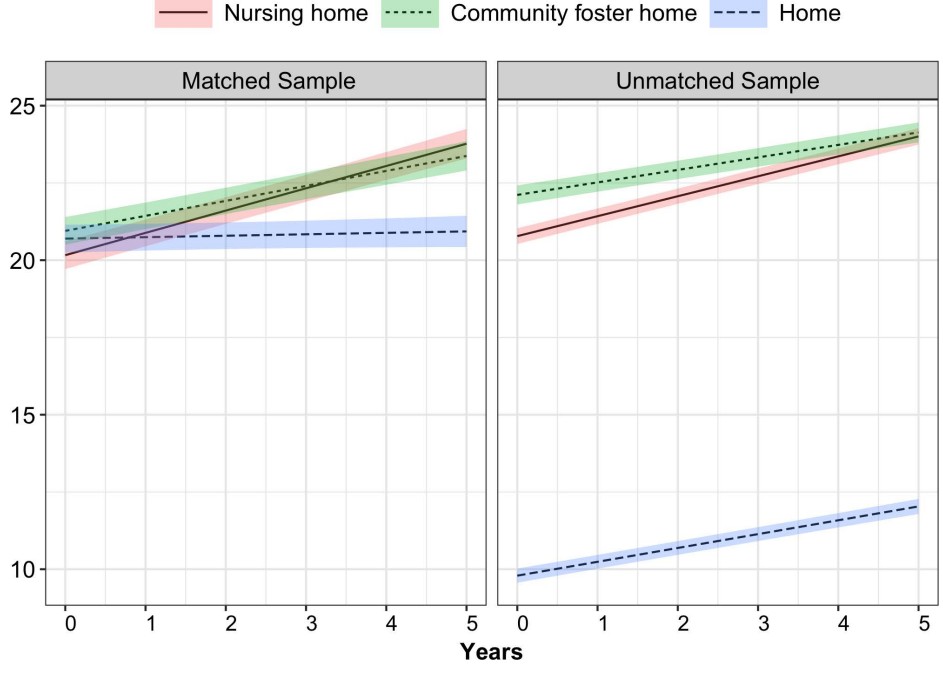

**Fig 1. Level of care scores by setting.**

**Table 3. Average scores for LOC items (mean ±SD) by setting at baseline in the matched samples.**

| | All | Home | Foster home | Nursing home | P |
|---|---|---|---|---|---|
| **Individual items (maximum allowable points)** | | | | | |
| III_Vision/Hearing/Speech (2) | 0.82±0.49 | 0.84±0.53 | 0.93±0.34 | 0.68±0.56 | <0.0001 |
| IV_Communication (2) | 0.71±0.61 | 0.81±0.63 | 0.77±0.55 | 0.56±0.62 | <0.0001 |
| V_Memory (2) | 1.42±0.71 | 1.43±0.72 | 1.52±0.61 | 1.30±0.76 | 0.0008 |
| VI_Mental Status/Behavior (9) | 1.67±1.66 | 1.72±1.96 | 1.78±1.42 | 1.49±1.54 | 0.0911 |
| VII_Feeding (2) | 0.88±0.58 | 0.87±0.61 | 0.85±0.53 | 0.92±0.58 | 0.291 |
| VIII_Transferring (4) | 2.78±0.84 | 2.70±0.95 | 2.62±0.82 | 3.02±0.69 | <0.0001 |
| IX_Mobility (5) | 3.85±1.13 | 3.77±1.14 | 3.76±1.15 | 4.00±1.09 | 0.0217 |
| X_Bowel Function/Continence (3) | 1.57±0.94 | 1.64±1.05 | 1.57±0.75 | 1.52±1.00 | 0.3139 |
| XI_Bladder Function/Continence (3) | 2.38±0.89 | 2.47±0.88 | 2.52±0.71 | 2.14±1.02 | <0.0001 |
| XII_Bathing (3) | 2.19±0.98 | 2.35±0.94 | 2.02±1.00 | 2.18±0.98 | 0.0003 |
| XIII_Dressing and Personal Grooming (3) | 2.24±0.55 | 2.34±0.59 | 2.20±0.57 | 2.18±0.46 | 0.0011 |
| **Aggregated items** | | | | | |
| Cognition (V-VI: 11 points) | 3.08±2.02 | 3.15±2.34 | 3.30±1.71 | 2.79±1.93 | 0.0079 |
| ADL (VII-XIII: 23 points) | 15.88±3.67 | 16.14±3.85 | 15.54±3.36 | 15.96±3.76 | 0.1365 |

Fig 2 presents the baseline LOC scores and changes in the scores for ADL and cognition separately by setting. For ADL, members in three settings had the same scores at baseline but the rate of increase was the largest among nursing home residents, followed by those living in foster homes. The LOC scores for ADL among individuals residing at home increased the slowest. For cognition, the differences of memory and mental status among setting are very small relative to differences in total cognitive score.

## Discussion

The predominant finding of this study is that, after matching for baseline LOC score and demographic characteristics, individuals in foster homes and nursing homes declined at a substantially higher rate than individuals at home among those with high functional needs. We found home care emerged as the best choice for maintaining one's functional status among the three settings, as the functional status of Medicaid members residing at home remained stable over time. Despite these benefits, individuals with greater functional needs were less likely to age-in-place in our sample compared to placement in foster homes or institutional settings. This research contributes to the broader literature on supports for dually eligible individuals, particularly those with concurrent disabilities who are eligible for Medicaid funded long-term services and support.

Compared with individuals residing at home in the unmatched sample, the matched sample tends to concentrate those with higher LOC scores living at home. This concentration possibly explains the observed discrepancy in the change of functional status among individuals residing at home in the matched versus unmatched samples. This finding speaks to the importance of early interventions especially considering greater effects of HCBS on health among younger individuals, those with less severe illnesses, and those with stronger social support [43]. Future research should focus on identifying factors within and beyond settings that contribute to the deterioration of health of individuals.

Our results also suggest that HCBS participants are not a homogeneous group and HCBS settings should be disaggregated and considered separately in future studies examining environmental or contextual factors on older adults' health. Individuals residing at home versus foster homes exhibited distinct patterns in functional status over time, despite both being considered HCBS settings. Specifically, foster home residents experienced notable decline over time, while the functional status of home residents was stable. Although a gap exists in the literature comparing the health effects of

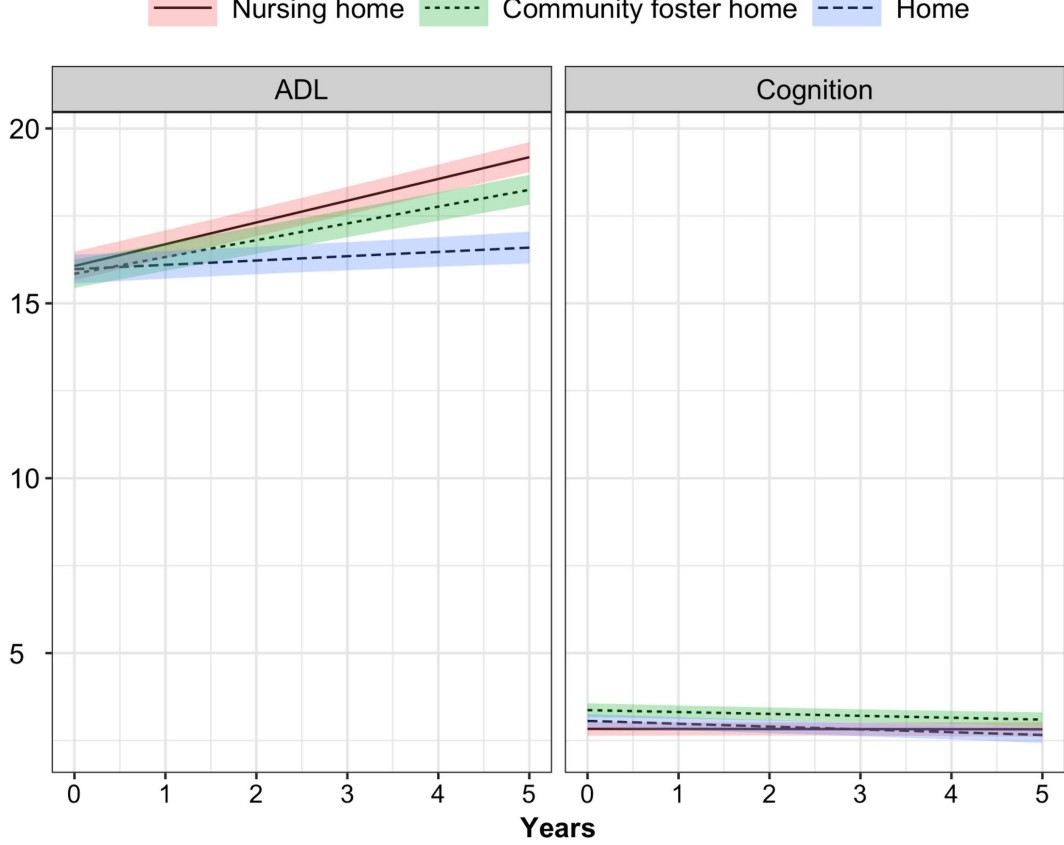

**Fig 2. Level of care scores for ADL and cognition by setting.**

home-based versus foster home care while considering selection bias, previous studies suggest setting-related factors may contribute to the poor outcomes often observed in community based foster home settings. For example, a 2009 study by the AARP Public Policy Institute found that adult foster homes frequently operate at a financial loss, with Medicaid reimbursement rates often failing to cover costs. They also found low availability of nursing support, although the purpose of HCBS is to provide medical care as needed in HCBS settings. The study also highlighted low training among caregivers [34]. Furthermore, there are also likely selection factors that impact outcomes over time, older adults with informal caregivers (e.g., family and friends) and stable housing are more likely to continue living at home as opposed to entering residential care [44–46]. Meanwhile, research has also shown a positive association between homeownership and health [47] and a positive link between having informal caregivers and functional independence among older adults [48]. These setting-related factors likely explain the different health effects between home-based care versus foster home care.

In fact, when compared to residents at home, those in foster homes exhibited a pattern of functional decline closely resembling that observed in nursing home residents. This finding adds to the scant literature on the health outcomes of foster home residents. Only one study focusing on veterans found that levels of comorbidity and frailty of individuals in the Medical Foster Home (MFH) program were comparable to those in nursing homes [49]. However, this study is descriptive and calls for future research addressing baseline imbalances before comparison [49]. Furthermore, as mentioned in the introduction, foster homes in the VA system may have different outcomes compared to those in Medicaid, as they operate under distinct funding streams and regulatory guidelines. The impact of these differences on quality of care, training,

funding, and their influence on older adults' health is a promising area for future HCBS research, as well as broader studies on residential treatment options for older adults.

While both foster home and nursing home residents undergo functional decline, a noteworthy observation is that the pace of deterioration appears to be more rapid among nursing home residents. Our finding is consistent with a small randomized controlled trial study by Oktay and Volland [50], which found that foster home residents were more likely to maintain or improve their ADL compared to nursing home residents. In contrast, a study conducted using 1988–1989 Oregon data found improved ADL functioning among nursing home residents but more functional decline among foster home residents over a 12-month period [51]. In our study, the slower rate of deterioration among foster home residents may be attributed to foster homes providing avenues for social activities and the development of close relationships [52], potentially enhancing the overall well-being and functioning of older adults. Research has shown that, after controlling for disability status, foster home residents reported more social activities compared to nursing home residents [53] and social activity is found to decrease the risk of long-term facility placement among community-dwelling older adults [54]. Qualitative research on MFH has identified that families and patients select them partly because they may offer a more home-like setting, and avoid them if they feel they may not be equipped to meet overall care needs [55]. Overall, research is scant on foster homes and their relative efficacy compared to private homes or nursing homes. This study helps to fill an important gap, as foster homes are increasingly identified as a community-based resource for individuals who lack natural support at home or financial resources, but who want to avoid institutional settings [56].

Additionally, our findings suggest the increase in care needs among individuals in nursing homes and foster homes was largely driven by functional deterioration rather than cognitive decline. In other words, the setting—the place where an individual lives—makes a larger difference in determining individuals' functionality compared to their cognition (i.e., memory, mental health/behavior). This is consistent with the literature. For example, Wang and Yang (2022) found that informal care from family members or friends, which is often provided at home, significantly slows down the of functional deterioration of care recipients, but not their depressive symptoms [57]. The dominating effect of ADL over cognition in our finding is also likely be attributed to the design of the LOC assessment form, which includes seven items with 23 maximum allowable points for ADL-related items (e.g., feeding and transferring) but only two items with 11 maximum allowable points for cognition-related items (i.e., memory and mental health/behavior). It is possible that the assessment tool does not adequately capture the cognition-related LOC care needs, such as support for issues with relationship due to cognitive limitations. Additionally, the "Mental health/behavior" item in the assessment form combines various aspects—such as orientation, aggressive behavior, self-injury, and wonder—into a single item. The grouping might make it challenging for assessors to accurately assign points that reflect an individual's cognition-related LOC care needs.

This study has a few limitations. First, we excluded those who switched between settings and those with only one LOC assessment from our analysis. Such exclusion may lead to bias, as individuals who stayed in the same setting may tend to have a more stable health status. In fact, our descriptive table (Table 1) indicated that the excluded individuals had higher LOC scores than those who stayed at home but lower scores than those who stayed in foster homes or nursing homes on average at baseline. This study also did not consider dropouts due to reasons such as death and relocation, which may introduce bias of our results. The data limitation prevents us from addressing or assessing this potential bias. These caveats should be considered when interpreting our findings, and future studies should address attrition when more comprehensive data are available. Second, this study does not exhaust all the factors that can contribute to the functional status of older adults, and therefore, we do not conclude causality of settings on health. For example, due to data limitation, this analysis does not include factors such as social connection and specific services provided that may partly explain the variation in functional decline. Third, despite employing various strategies and stringent matching criteria, perfect matching was not achieved with residual imbalance observed in variables such as race/ethnicity and baseline LOC scores. To mitigate the potential impact of the imbalance and enhance the robustness of our findings, we included baseline LOC scores and race/ethnicity as covariates in the regression models. Lastly, the analysis based on a sample of

Medicaid recipients in Hawaii may limit the generalizability of our findings. However, this study is valuable as Hawaii is a state with large Asian and Pacific Islander populations, who are often underrepresented in research.

## Conclusion

This study examined the change in the functional status of dually eligible Medicaid recipients aged 65 or older in Hawaii across three settings: home, foster home, and nursing home. Using matching and linear mixed-effects modeling, we identified distinct patterns in the change of functional status across settings. While residents at home maintained stable functioning, those in foster homes and nursing homes exhibited functional declines over time with a more rapid rate of decline among nursing home residents, closely matched within foster homes. We also identified different patterns among residents at home in the unmatched versus matched samples, as individuals residing at home with higher functional needs tend to concentrate in the matched sample.

This study significantly contributes to the HCBS literature, and literature on long-term care for dually eligible individuals with disabilities by comparing functional outcomes over time across three settings. In addition, we addressed the baseline differences in individuals' characteristics for enhanced comparability across settings. The findings also underscore the need to consider the heterogeneity within HCBS settings for targeted interventions to improve health. This study also adds to the scant literature on the health outcomes of foster home care, and suggests that foster home selection factors and quality should be closely considered by state Medicaid agencies using foster homes as an HCBS housing option.

This study suggests, from the perspective of maintaining overall functional stability, promoting home-based care as a viable and sustainable alternative to institutional care. This should be accompanied with adequate support for informal caregivers given the negative consequences of the caregiving burden on caregivers' health and well-being [58–61]. Note that this study focuses on functional status. Understanding other health outcomes—such as mortality, hospitalization, and quality of life—in various settings can be as equally critical in determining the most suitable care options. Future studies should explore the extent to which these health outcomes vary across care settings, as such knowledge could inform decision about effective care strategies to enhance overall health.

For foster homes and nursing homes, it is imperative to identify factors that mitigate or contribute to functional deterioration. Additionally, recognizing distinct patterns in matched versus unmatched samples highlights the urgency of early interventions. Programs like Hawaii's at-risk initiative, offering HCBS to those at risk of institutional level of care, have the potential to delay health deterioration. Meanwhile, it is crucial to equip families and caregivers with an understanding of the disparate functional outcomes across settings. This knowledge informs decisions on care setting choices, considering factors such as the rate of functional decline. Emphasizing education empowers families to navigate care options effectively and advocate for the well-being of their loved ones.

## Supporting information

**S1 Fig.  Sample selection process.**
(DOCX)

## Author contributions

**Conceptualization:** Hua Zan, Yanyan Wu, Joy Agner.

**Data curation:** Hua Zan, Yanyan Wu.

**Formal analysis:** Yanyan Wu.

**Funding acquisition:** John P. Barile, Joshua R. Holmes.

**Investigation:** Hua Zan, Yanyan Wu, Yan Luo, John P. Barile, Joshua R. Holmes, Joy Agner.

**Methodology:** Hua Zan, Yanyan Wu, Joy Agner.

**Project administration:** John P. Barile.

**Resources:** John P. Barile, Joshua R. Holmes.

**Supervision:** Hua Zan, Joy Agner.

**Validation:** Hua Zan, Yanyan Wu, Yan Luo, John P. Barile, Joshua R. Holmes, Joy Agner.

**Writing – original draft:** Hua Zan, Yan Luo.

**Writing – review & editing:** Hua Zan, Yanyan Wu, Yan Luo, John P. Barile, Joshua R. Holmes, Joy Agner.

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
