## [Decision Letter · Decision Letter 0]

Dear Dr. Zan,

We look forward to receiving your revised manuscript.

Kind regards,

De-Chih Lee, Ph.D.

Academic Editor

PLOS ONE

Journal Requirements:

2. Thank you for stating the following financial disclosure: “This work was supported by Hawaii Department of Human Services, United States of America.”

Please state what role the funders took in the study.  If the funders had no role, please state: “The funders had no role in study design, data collection and analysis, decision to publish, or preparation of the manuscript.”

5. We notice that your supplementary figure are included in the manuscript file. Please remove them and upload them with the file type 'Supporting Information'. Please ensure that each Supporting Information file has a legend listed in the manuscript after the references list.

Additional Editor Comments:

Please make a major revision based on the reviewer's comments.

Reviewers' comments:

Reviewer's Responses to Questions

**Comments to the Author**

1. Is the manuscript technically sound, and do the data support the conclusions?

Reviewer #1: No

Reviewer #2: Yes

2. Has the statistical analysis been performed appropriately and rigorously?

Reviewer #1: No

Reviewer #2: Yes

3. Have the authors made all data underlying the findings in their manuscript fully available?

Reviewer #1: Yes

Reviewer #2: Yes

4. Is the manuscript presented in an intelligible fashion and written in standard English?

Reviewer #1: Yes

Reviewer #2: Yes

Reviewer #1: Major Objections

1. The scale used is specific to Hawaii and is not sufficiently described in the article to ascertain its effectiveness in evaluating the functional status of participants. One must refer to the references to understand this scale, which appears to evaluate both functional status (several questions similar to the Katz or Lawton scales) and cognitive status (questions V and VI). The authors explain that a total score out of 38 is obtained, but it is unclear how this score is calculated from the 17 questions on the scale. This aspect is crucial as placement in nursing homes typically results from significant cognitive deterioration rather than functional decline. It would be useful to reanalyze the data using a score based solely on questions VII to XVII, which are comparable to functional status scales commonly used in the scientific literature. Additionally, the cognitive scores (questions V and VI) for the matched sample between the three groups should be described. The goal is to understand whether the accelerated decline observed in nursing homes is due to cognitive or functional deterioration of the participants.

2. The exclusion of participants who changed residences is a major bias; at the very least, the number of such individuals should be described for the "home" group. If only participants who did not die during follow-up and did not change residences are compared, and there is no adjustment for cognitive status, it is unsurprising that the total score trajectory is unfavorable for nursing homes. A competitive risk model including mortality could have been used for this study. All these aspects greatly limit the interpretation of the results, despite efforts made using the propensity score.

Minor Objections

3. The Hawaiian foster home model proposed should be better described in terms of criteria for admission, medical and paramedical supervision, and social interactions. Is this model intended for frail individuals, and if so, based on what criteria?

4. Abstract: The summary should be revised to specify the sample sizes, methods, and main quantified results. A single concluding sentence is sufficient.

5. Introduction: There are several contradictory passages, for example regarding mortality: “Studies comparing residents living in assisted living versus nursing homes also did not find a significant difference in physical function, cognition, mental health, or mortality outcomes between the two settings over time.” And immediately after: “For example, nursing home residents tend to have higher risk of mortality, significant comorbidities and more complex needs, lower functional status, and use more intensive health care services compared to individuals living in HCB settings.”

6. The paragraph on the “Conceptual Framework” is not very useful and could be summarized in two or three sentences.

7. The hyperlink “Form 1147” does not work.

8. The title of Table 2 should be more explanatory.

9. Discussion: I do not understand the sentence: “This observation aligns with findings from previous studies. For example, an Oregon study identified high substitutability of adult foster care for nursing home care. That is, with every additional foster home resident in a county, a nursing home in that county loses a resident.”

10. The limits of the study should be included in the discussion section mais should be developed (see major comments).

Reviewer #2: This study compares functional decline over time in 3 settings: home, foster homes and nursing home.

This article provides relevant information to the readers. I praise the authors for their research.

While authors highlight differences in age, race, marital status and setting of care, I believe authors could highlight the differences in comorbidities, cognitive impairment between the individuals selected in each setting. It is only intuitive to think that individual are admitted to the nursing home with higher level of medical complexity, comorbidities or cognitive decline that make them require more assistance in ADLs. I would like to see how these characteristics pan out as comparisons between each group, as they may determine the degree of decline that is being identified. The degree of decline might be associated with comorbidities and medical conditions rather than actual settings alone. I feel that comparisons of degree of decline will be stronger if these are incorporated into the analysis.

In the discussion there is mention of 'better health' among individuals who receive care at home. Could you expand on what specifically are you referring to by 'better health'?

**Do you want your identity to be public for this peer review?** For information about this choice, including consent withdrawal, please see our Privacy Policy

Reviewer #1: **Yes: ** Denis Boucaud-Maitre

Reviewer #2: No

---

## [Author Response · Author response to Decision Letter 1]

2 Dec 2024

Responses to the editor and reviewers:

We appreciate the opportunity to revise and resubmit this manuscript and thank the reviewers for the insightful comments. We have revised the manuscript according to reviewers' constructive feedback. Our specific responses to each comment are below.

Journal Requirements:

Comment 1. When submitting your revision, we need you to address these additional requirements.

RESPONSE:

We have revised the manuscript following style guidelines.

Comment 2. Thank you for stating the following financial disclosure: “This work was supported by Hawaii Department of Human Services, United States of America.”

Please state what role the funders took in the study. If the funders had no role, please state: “The funders had no role in study design, data collection and analysis, decision to publish, or preparation of the manuscript.”

RESPONSE:

We have updated the funder statement in the title page and also included it in the cover letter.

Comment 3. We note that you have indicated that there are restrictions to data sharing for this study. For studies involving human research participant data or other sensitive data, we encourage authors to share de-identified or anonymized data. However, when data cannot be publicly shared for ethical reasons, we allow authors to make their data sets available upon request. For information on unacceptable data access restrictions, please see http://journals.plos.org/plosone/s/data-availability#loc-unacceptable-data-access-restrictions.

RESPONSE:

We have updated the Data Availability Statement in the title page.

Comment 4. Please include your full ethics statement in the ‘Methods’ section of your manuscript file. In your statement, please include the full name of the IRB or ethics committee who approved or waived your study, as well as whether or not you obtained informed written or verbal consent. If consent was waived for your study, please include this information in your statement as well.

RESPONSE:

We have included full ethics statements in the Method section.

Comment 5. We notice that your supplementary figure are included in the manuscript file. Please remove them and upload them with the file type 'Supporting Information'. Please ensure that each Supporting Information file has a legend listed in the manuscript after the references list.

RESPONSE:

We have revised accordingly.

Additional Editor Comments:

Please make a major revision based on the reviewer's comments.

Reviewers' comments:

Reviewer's Responses to Questions

Comments to the Author

1. Is the manuscript technically sound, and do the data support the conclusions?

Reviewer #1: No

Reviewer #2: Yes

2. Has the statistical analysis been performed appropriately and rigorously?

Reviewer #1: No

Reviewer #2: Yes

RESPONSE TO QUESTION 1 AND 2:

We have included additional analyses and revised the manuscript as suggested. Please see the detailed responses to specific comments below.

3. Have the authors made all data underlying the findings in their manuscript fully available?

Reviewer #1: Yes

Reviewer #2: Yes

4. Is the manuscript presented in an intelligible fashion and written in standard English?

Reviewer #1: Yes

Reviewer #2: Yes

5. Review Comments to the Author

Reviewer #1: Major Objections

1. The scale used is specific to Hawaii and is not sufficiently described in the article to ascertain its effectiveness in evaluating the functional status of participants. One must refer to the references to understand this scale, which appears to evaluate both functional status (several questions similar to the Katz or Lawton scales) and cognitive status (questions V and VI). The authors explain that a total score out of 38 is obtained, but it is unclear how this score is calculated from the 17 questions on the scale. This aspect is crucial as placement in nursing homes typically results from significant cognitive deterioration rather than functional decline. It would be useful to reanalyze the data using a score based solely on questions VII to XVII, which are comparable to functional status scales commonly used in the scientific literature. Additionally, the cognitive scores (questions V and VI) for the matched sample between the three groups should be described. The goal is to understand whether the accelerated decline observed in nursing homes is due to cognitive or functional deterioration of the participants.

RESPONSE:

Thank you to the reviewer for the suggestions. We conducted the following additional analyses and updated the paper accordingly:

(1) providing summary statistics of each LOC item in the assessment form by setting after matching and including maximum points for each item (Table 3)

(2) tracking the changes in LOC scores for ADL-related items (items VII to XVII) by setting and the changes in LOC scores for cognition-related items (items V and VI) by setting (Fig 2)

2. The exclusion of participants who changed residences is a major bias; at the very least, the number of such individuals should be described for the "home" group. If only participants who did not die during follow-up and did not change residences are compared, and there is no adjustment for cognitive status, it is unsurprising that the total score trajectory is unfavorable for nursing homes. A competitive risk model including mortality could have been used for this study. All these aspects greatly limit the interpretation of the results, despite efforts made using the propensity score.

RESPONSE:

We agree with the reviewer about these limitations. To address the limitation, we compared the baseline characteristics of individuals excluded from the analysis with those who remained in the same settings for 2+ years before matching in Table 1. The differences across these groups were highlighted in the manuscript (page 5, data and sample section). Unfortunately, we were unable to incorporate mortality in the modeling due to the data limitations. We emphasized these caveats should be considered when interpreting our findings in the limitation section.

Minor Objections

3. The Hawaiian foster home model proposed should be better described in terms of criteria for admission, medical and paramedical supervision, and social interactions. Is this model intended for frail individuals, and if so, based on what criteria?

RESPONSE:

Hawaii has administrative rules which establish comprehensive guidelines for the certification, administration, and care standards of adult foster home. These rules specify care protocols for residents and emphasize their rights including dignity and participation in community life. Despite the detailed rules, few local or national studies have been conducted about how adult foster homes actually run, health outcomes of residents, and differences compared to nursing homes.

We believe one of the key contributions of this study is its addition to the literature on the relative impact of adult foster care compared to private residences and nursing homes. We agree with the reviewer that the findings related to the nursing homes are relatively unsurprising given prior research showing faster functional decline among nursing home residents compared to those in home settings. However, we did not anticipate that residents in foster homes would experience a similar rate of decline compared to nursing homes. Foster homes and private residences are typically lumped together as "home and community-based settings." Therefore, we see this finding regarding the decline in foster care settings as particularly significant. We hope it encourages further investigation and research on foster care settings, which are distinct from other forms of adult residential treatment, such as assisted living facilities.

4. Abstract: The summary should be revised to specify the sample sizes, methods, and main quantified results. A single concluding sentence is sufficient.

RESPONSE:

We have revised the abstract as suggested. Here is the relevant revised text: “Among 5,315 older adult Medicaid recipients, we found distinct characteristics in initial placement. Individuals placed at home were younger and had lower functional impairment scores compared to individuals in foster homes or nursing homes. To increase comparability despite these differences, we matched older adults (n=852) on baseline functional status, age, sex, marital status, and race/ethnicity using propensity score matching and performed sensitivity analyses on cognitive status. After matching, linear mixed-effects modeling revealed a notably slower rate of functional decline at home compared to nursing homes or foster homes. Individuals at home had fairly stable functional status (low deterioration) over the eight years. Nursing home residents had the fastest rate of decline, followed closely by individuals in foster homes. These findings suggest that, when possible, Medicaid agencies should prioritize supporting long-term care in private homes.”.

5. Introduction: There are several contradictory passages, for example regarding mortality: “Studies comparing residents living in assisted living versus nursing homes also did not find a significant difference in physical function, cognition, mental health, or mortality outcomes between the two settings over time.” And immediately after: “For example, nursing home residents tend to have higher risk of mortality, significant comorbidities and more complex needs, lower functional status, and use more intensive health care services compared to individuals living in HCB settings.”

RESPONSE:

Thanks for pointing this out. We have revised the corresponding section.

6. The paragraph on the “Conceptual Framework” is not very useful and could be summarized in two or three sentences.

RESPONSE:

We have revised as suggested.

7. The hyperlink “Form 1147” does not work.

RESPONSE:

We have updated the link.

8. The title of Table 2 should be more explanatory.

RESPONSE:

We have revised as suggested.

9. Discussion: I do not understand the sentence: “This observation aligns with findings from previous studies. For example, an Oregon study identified high substitutability of adult foster care for nursing home care. That is, with every additional foster home resident in a county, a nursing home in that county loses a resident.”

RESPONSE:

We have removed this sentence and emphasized our contribution to the scant literature on the health outcomes of foster home residents.

10. The limits of the study should be included in the discussion section mais should be developed (see major comments).

RESPONSE:

We have revised as suggested.

Reviewer #2: This study compares functional decline over time in 3 settings: home, foster homes and nursing home.

This article provides relevant information to the readers. I praise the authors for their research.

While authors highlight differences in age, race, marital status and setting of care, I believe authors could highlight the differences in comorbidities, cognitive impairment between the individuals selected in each setting. It is only intuitive to think that individual are admitted to the nursing home with higher level of medical complexity, comorbidities or cognitive decline that make them require more assistance in ADLs. I would like to see how these characteristics pan out as comparisons between each group, as they may determine the degree of decline that is being identified. The degree of decline might be associated with comorbidities and medical conditions rather than actual settings alone. I feel that comparisons of degree of decline will be stronger if these are incorporated into the analysis.

RESPONSE:

The assessment data include the text entries for the primary and secondary diagnoses, and assessors can type in multiple diagnoses for each individual. This data format and quality make it challenging to efficiently extract diagnosis information. However, we focused on two types of primary/secondary diagnoses—mental illnesses and dementia—and compared the baseline characteristics by diagnosis and setting. Specifically, we used key terms including "schiz”, “bipol”, “depress”, “psycho”, “dementia”, and “alzhem” (in both lower and upper case) to identify mental illnesses, and “dementia” or “alzhem” to identify dementia. In both unmatched and matched sample, we identified higher percentages of nursing home residents with mental illnesses or dementia compared to those living at home or in nursing homes.

Table. Baseline Characteristics by Diagnosis in the Unmatched and Matched Samples of Individuals Staying in the Same Setting with ≥ 2-Year of Follow-up

Primary/secondary diagnosis Unmatched sample Matched sample

Home Foster home Nursing home Home Foster home Nursing home

Mental illness Yes 191 (8.1%) 39 (3.3%) 322 (18.1%) 19 (6.7%) 11 (3.9%) 60 (21.1%)

No 2,163 (91.9%) 1,14

---

## [Decision Letter · Decision Letter 1]

Dear Dr. Zan,

Thank you for submitting your manuscript to PLOS ONE. After careful consideration, we feel that it has merit but does not fully meet PLOS ONE’s publication criteria as it currently stands. Therefore, we invite you to submit a revised version of the manuscript that addresses the points raised during the review process.

We look forward to receiving your revised manuscript.

Kind regards,

M. Mahmud Khan

Academic Editor

PLOS ONE

Journal Requirements:

Reviewers' comments:

Reviewer's Responses to Questions

**Comments to the Author**

Reviewer #1: All comments have been addressed

2. Is the manuscript technically sound, and do the data support the conclusions?

Reviewer #1: Partly

3. Has the statistical analysis been performed appropriately and rigorously?

Reviewer #1: Yes

4. Have the authors made all data underlying the findings in their manuscript fully available?

Reviewer #1: Yes

5. Is the manuscript presented in an intelligible fashion and written in standard English?

Reviewer #1: Yes

Reviewer #1: The article has been substantially revised, incorporating differentiated analyses on cognition and functional status. The main finding is that residents in foster homes exhibited a greater functional decline compared to those residing in private homes.

I maintain that the conclusions and interpretations of the results should be approached with greater nuance, particularly regarding the statement: “These findings suggest that, when possible, Medicaid agencies should prioritize supporting long-term care in private homes.” Placement in nursing homes or foster homes occurs when remaining at home is either undesirable or unfeasible.

Comorbidities such as diabetes, renal insufficiency, stroke history, hemiplegia, cancer, etc., which may explain placement in nursing or foster homes, were not accounted for. While a decline in functional status is indeed observed, other parameters are equally important, including mortality, hospitalizations, quality of life, social isolation etc. These factors play a critical role in guiding policymakers and families in choosing among various housing options. It is therefore, in my opinion, inappropriate to contrast these alternatives based solely on functional status.

**Do you want your identity to be public for this peer review?** For information about this choice, including consent withdrawal, please see our Privacy Policy

Reviewer #1: **Yes: ** BOUCAUD-MAITRE

---

## [Author Response · Author response to Decision Letter 2]

15 Jan 2025

We appreciate the opportunity to revise and resubmit this manuscript and thank the reviewer for the insightful comments. We have revised the manuscript according to reviewers' constructive feedback. Our specific responses to each comment are below.

Comments to the Author

1. If the authors have adequately addressed your comments raised in a previous round of review and you feel that this manuscript is now acceptable for publication, you may indicate that here to bypass the “Comments to the Author” section, enter your conflict of interest statement in the “Confidential to Editor” section, and submit your "Accept" recommendation.

Reviewer #1: All comments have been addressed

2. Is the manuscript technically sound, and do the data support the conclusions?

Reviewer #1: Partly

RESPONSE:

We have revised the abstract and conclusion sections to ensure alignment between the data and the conclusions.

3. Has the statistical analysis been performed appropriately and rigorously?

Reviewer #1: Yes

4. Have the authors made all data underlying the findings in their manuscript fully available?

Reviewer #1: Yes

5. Is the manuscript presented in an intelligible fashion and written in standard English?

Reviewer #1: Yes

6. Review Comments to the Author

Reviewer #1: The article has been substantially revised, incorporating differentiated analyses on cognition and functional status. The main finding is that residents in foster homes exhibited a greater functional decline compared to those residing in private homes.

I maintain that the conclusions and interpretations of the results should be approached with greater nuance, particularly regarding the statement: “These findings suggest that, when possible, Medicaid agencies should prioritize supporting long-term care in private homes.” Placement in nursing homes or foster homes occurs when remaining at home is either undesirable or unfeasible.

Comorbidities such as diabetes, renal insufficiency, stroke history, hemiplegia, cancer, etc., which may explain placement in nursing or foster homes, were not accounted for. While a decline in functional status is indeed observed, other parameters are equally important, including mortality, hospitalizations, quality of life, social isolation etc. These factors play a critical role in guiding policymakers and families in choosing among various housing options. It is therefore, in my opinion, inappropriate to contrast these alternatives based solely on functional status.

RESPONSE:

We have revised the abstract and conclusion sections, highlighting the study's focus on functional status in the conclusion. Further research is needed to explore other health outcomes, such as mortality, hospitalization, and quality of life across care settings, to better inform effective care options for improving health.

7. PLOS authors have the option to publish the peer review history of their article (what does this mean?). If published, this will include your full peer review and any attached files.

Do you want your identity to be public for this peer review? For information about this choice, including consent withdrawal, please see our Privacy Policy.

Reviewer #1: Yes: BOUCAUD-MAITRE

---

## [Decision Letter · Decision Letter 2]

PLOS ONE

Dear Dr. Zan,

Thank you for submitting your manuscript to PLOS ONE. After careful consideration, we feel that it has merit but does not fully meet PLOS ONE’s publication criteria as it currently stands. Therefore, we invite you to submit a revised version of the manuscript that addresses the points raised during the review process.

Overall, the clarity of certain sections of the manuscript, particularly the Methods section, require improvement to enhance readers' understanding of the analyses performed and to ensure reproducibility. For instance, the following statement lacks clarity: “Individuals at home were matched to those living in foster homes and nursing homes respectively, and then the two separate matched samples were merged by individuals in the home setting." Specifically, the meaning of “merged by individuals in the home setting” is unclear and needs further explanation. 

It is also pertinent to explicitly include the regression equations used for the analysis to provide transparency and facilitate reproducibility. Furthermore, illustrating the estimated slopes using figures would significantly improve the presentation and understanding of results. For reference, the figures presented in the article titled “Poor hemorrhagic stroke outcomes during the COVID-19 pandemic are driven by socioeconomic disparities: analysis of nationally representative data” (BMJ Neurology Open) offer a good example of how to visualize equations involving change in slope.

Additionally, clarification is required for the following statement: “In addition, since the assessment form covers various aspects of LOC needs—such as ADL, which are comparable to functional status scales in the literature, and cognition—5 we further compared the average scores for each assessment item by setting after matching and analyzed the change in LOC scores for ADL (Items VII-XIII in Form 1147) and cognition (Items V-VI) separately by setting.” The phrasing is presently ambiguous, particularly regarding the comparison and analysis of LOC scores. A more precise description would greatly enhance the reader’s comprehension.

While the manuscript attempts to justify why race and LOC remained statistically significant despite matching, such results often point to imperfect matching and residual confounding, which could bias the conclusions. To address this, I strongly recommend evaluating and reporting balance metrics for the propensity score model, utilizing tools such as standardized mean differences or visual plots to assess covariate balance. Additionally, exploring alternative methods, including propensity score stratification or inverse probability weighting (IPW), may help to address the issue of imperfect balance and improve the robustness of the findings.

Finally, it is concerning that the authors evaluated the Medicaid population aged 65 and older without addressing the issue of dual eligibles or discussing how their dual status might influence the results. It is equally important to provide context on how these older adults came to be Medicaid recipients. Such omissions leave critical gaps in the interpretation of the findings and the broader implications of the study. I strongly recommend that the authors include a discussion of dual eligibility and its potential impact, as well as a clear explanation of the circumstances leading to older adults being enrolled in Medicaid.

We look forward to receiving your revised manuscript.

Kind regards,

Abdulaziz T. Bako, MBBS; MPH; PhD

Academic Editor

PLOS ONE

Journal Requirements:

Reviewers' comments:

Reviewer's Responses to Questions

**Comments to the Author**

Reviewer #1: All comments have been addressed

2. Is the manuscript technically sound, and do the data support the conclusions?

Reviewer #1: Yes

3. Has the statistical analysis been performed appropriately and rigorously?

Reviewer #1: Yes

4. Have the authors made all data underlying the findings in their manuscript fully available?

Reviewer #1: Yes

5. Is the manuscript presented in an intelligible fashion and written in standard English?

Reviewer #1: Yes

Reviewer #1: Dear authors,

thank you for taking into account my latest comments, which help to qualify your results, and congratulations on your article.

**Do you want your identity to be public for this peer review?** For information about this choice, including consent withdrawal, please see our Privacy Policy

Reviewer #1: **Yes: ** Boucaud-Maitre

---

## [Author Response · Author response to Decision Letter 3]

14 May 2025

Responses to the editor and the reviewer:

We appreciate the opportunity to revise and resubmit this manuscript and thank you for the insightful comments. We have revised the manuscript according to the constructive feedback. Our specific responses to each comment are below.

PONE-D-24-22552R2

Home settings are associated with less functional decline among older adults compared to community-care foster homes and skilled nursing facilities in Hawaii

PLOS ONE

Dear Dr. Zan,

Thank you for submitting your manuscript to PLOS ONE. After careful consideration, we feel that it has merit but does not fully meet PLOS ONE’s publication criteria as it currently stands. Therefore, we invite you to submit a revised version of the manuscript that addresses the points raised during the review process.

1. Overall, the clarity of certain sections of the manuscript, particularly the Methods section, require improvement to enhance readers' understanding of the analyses performed and to ensure reproducibility. For instance, the following statement lacks clarity: “Individuals at home were matched to those living in foster homes and nursing homes respectively, and then the two separate matched samples were merged by individuals in the home setting." Specifically, the meaning of “merged by individuals in the home setting” is unclear and needs further explanation.

RESPONSE:

We have revised to increase clarity of the manuscript especially the Methods section. For example, we revised “Individuals at home were matched to those living in foster homes and nursing homes respectively, and then the two separate matched samples were merged by individuals in the home setting” to “Because the matching method can only match two groups at a time, we first matched individuals at home to those living in foster homes and nursing homes respectively. Then, the two separate matched samples were linked by the individuals in the home setting, ensuring that all three settings were matched.”.

2. It is also pertinent to explicitly include the regression equations used for the analysis to provide transparency and facilitate reproducibility.

RESPONSE:

Good suggestion. We added the regression equation to the manuscript.

3. Furthermore, illustrating the estimated slopes using figures would significantly improve the presentation and understanding of results. For reference, the figures presented in the article titled “Poor hemorrhagic stroke outcomes during the COVID-19 pandemic are driven by socioeconomic disparities: analysis of nationally representative data” (BMJ Neurology Open) offer a good example of how to visualize equations involving change in slope.

RESPONSE:

Thank you for highlighting the illustration method used in the article title “Poor hemorrhagic stroke outcomes during the COVID-19 pandemic are driven by socioeconomic disparities: analysis of nationally representative data” (hereafter referred as the BMJ article). Upon closer examination, we have noted that the BMJ article and our current work addressed distinct research questions, which led us to use different analytical approaches and visualization techniques.

The BMJ article's objective was to determine the impact of the COVID-19 pandemic on intracerebral hemorrhage mortality, effectively visualized in their Figure 1 by illustrating the change in the slope of mortality trends between the pre-pandemic and pandemic periods using logistic regression models.

In contrast, our study focuses on comparing functionality, as measured by Level of Care (LOC) scores, across three different settings—home, foster home, and nursing home—over time. To achieve this, we utilized linear mixed-effects (LME) models. Our primary goal is to estimate the average baseline LOC scores for each setting (represented by the fixed-effects intercepts in our LME models) and to quantify the change in LOC scores over time within each setting (captured by the fixed-effect slopes). We believe that Figure 1 in our manuscript effectively illustrates these key findings.

4. Additionally, clarification is required for the following statement: “In addition, since the assessment form covers various aspects of LOC needs—such as ADL, which are comparable to functional status scales in the literature, and cognition—5 we further compared the average scores for each assessment item by setting after matching and analyzed the change in LOC scores for ADL (Items VII-XIII in Form 1147) and cognition (Items V-VI) separately by setting.” The phrasing is presently ambiguous, particularly regarding the comparison and analysis of LOC scores. A more precise description would greatly enhance the reader’s comprehension.

RESPONSE:

We revised and changed it to the following:

“The assessment form covers various aspects of LOC needs such as communication, memory, mobility, and dressing and grooming. To determine how these needs vary across settings, we compared the average scores for each assessment item by setting after matching. Given the established link between cognitive decline and increased care needs and worsened health outcomes (Connell et al., 2001; Millán-Calenti et al., 2013), we further aggregated cognition items (Items V-VI in Form 1147) and ADL items (Items VII-XIII in Form 1147). By comparing the scores of these two broad categories of care needs across settings, we aim to determine the influence of the change in cognition and functional status on shifts in LOC scores within the three settings.”

Reference:

Connell, C. M., Janevic, M. R., & Gallant, M. P. (2001). The costs of caring: impact of dementia on family caregivers. Journal of geriatric psychiatry and neurology, 14(4), 179-187.

Millán-Calenti, J. C., Tubío, J., Pita-Fernández, S., Rochette, S., Lorenzo, T., & Maseda, A. (2012). Cognitive impairment as predictor of functional dependence in an elderly sample. Archives of gerontology and geriatrics, 54(1), 197-201.

5. While the manuscript attempts to justify why race and LOC remained statistically significant despite matching, such results often point to imperfect matching and residual confounding, which could bias the conclusions. To address this, I strongly recommend evaluating and reporting balance metrics for the propensity score model, utilizing tools such as standardized mean differences or visual plots to assess covariate balance. Additionally, exploring alternative methods, including propensity score stratification or inverse probability weighting (IPW), may help to address the issue of imperfect balance and improve the robustness of the findings.

RESPONSE:

Thank you for the suggestion. To ensure an adequate sample size and maximize statistical power for detecting group differences, we employed various strategies and implemented stringent matching criteria. While the matching was not perfect, we included the matching variables as covariates in the regression models to mitigate the impact of any residual imbalance and enhance the robustness of our findings. We also acknowledged this limitation regarding imperfect balance in the discussion section.

6. Finally, it is concerning that the authors evaluated the Medicaid population aged 65 and older without addressing the issue of dual eligibles or discussing how their dual status might influence the results.

It is equally important to provide context on how these older adults came to be Medicaid recipients. Such omissions leave critical gaps in the interpretation of the findings and the broader implications of the study. I strongly recommend that the authors include a discussion of dual eligibility and its potential impact, as well as a clear explanation of the circumstances leading to older adults being enrolled in Medicaid.

RESPONSE:

Thank you for these comments. Because our sample was limited to Medicaid members over 65, they are all eligible for Medicare, all receiving Medicaid, and thus are all dually eligible. Since all participants are dually eligible, this would not affect the results of our analysis, which aimed to determine how delivery of HCBS services, specifically living and treatment context (home, foster home, or nursing facility) impacted functional outcomes over time. However, we see your point that members' dual eligibility status is important contextual information for the introduction and discussion.

Dual eligibles tend to have worse health outcomes, higher social needs, higher healthcare costs, racial disparities, and have less economic resources compared to non-Medicaid receiving older adults. Furthermore, individuals within our sample face additional challenges and vulnerabilities (even compared to other dual eligibles) as they have concurrent disability (defined by functional impairment) which makes them eligible for HCBS.

This constellation of social and medical risk factors are exactly the complex health challenges that HCBS intend to address, and framing within this social economic context is important. As for how individuals become eligible for Medicaid, we have included further details on eligibility for Medicaid. We added:

“To qualify for Medicaid in Hawaii, applicants must be state residents, meet specific citizenship or immigration requirements, and demonstrate limited financial resources. Income and asset limitations vary depending on the specific Medicaid program. In addition, accessing long-term care services and supports, such as HCBS and nursing home care, requires an assessment to confirm the functional need for such care. Upon approval following this evaluation, individuals gain access to tailored services and supports.”

In summary, we have added more discussion of challenges common to dual eligibles, added dual eligible as a keyword, and framed these findings as contributing additionally to research on services for dual eligibles receiving HCBS. Thank you for bringing this to our attention.

We look forward to receiving your revised manuscript.

Kind regards,

Abdulaziz T. Bako, MBBS; MPH; PhD

Academic Editor

PLOS ONE

Journal Requirements:

Reviewers' comments:

Reviewer's Responses to Questions

Comments to the Author

1. If the authors have adequately addressed your comments raised in a previous round of review and you feel that this manuscript is now acceptable for publication, you may indicate that here to bypass the “Comments to the Author” section, enter your conflict of interest statement in the “Confidential to Editor” section, and submit your "Accept" recommendation.

Reviewer #1: All comments have been addressed

2. Is the manuscript technically sound, and do the data support the conclusions?

Reviewer #1: Yes

3. Has the statistical analysis been performed appropriately and rigorously?

Reviewer #1: Yes

4. Have the authors made all data underlying the findings in their manuscript fully available?

Reviewer #1: Yes

5. Is the manuscript presented in an intelligible fashion and written in standard English?

Reviewer #1: Yes

6. Review Comments to the Author

Reviewer #1: Dear authors,

thank you for taking into account my latest comments, which help to qualify your results, and congratulations on your article.

7. PLOS authors have the option to publish the peer review history of their article (what does this mean?). If published, this will include your full peer review and any attached files.

Do you want your identity to be public for this peer review? For information about this choice, including consent withdrawal, please see our Privacy Policy.

Reviewer #1: Yes: Boucaud-Maitre

---

## [Editor Report · Decision Letter 3]

Home settings are associated with less functional decline among older adults compared to community-care foster homes and skilled nursing facilities in Hawaii

PONE-D-24-22552R3

Dear Dr. Zan,

We’re pleased to inform you that your manuscript has been judged scientifically suitable for publication and will be formally accepted for publication once it meets all outstanding technical requirements.

Kind regards,

Abdulaziz T. Bako, MBBS; MPH; PhD

Academic Editor

PLOS ONE
---

## [Editor Report · Acceptance letter]

PONE-D-24-22552R3

PLOS ONE

Dear Dr. Zan,

I'm pleased to inform you that your manuscript has been deemed suitable for publication in PLOS ONE. Congratulations! Your manuscript is now being handed over to our production team.

Kind regards,

on behalf of

Dr. Abdulaziz T. Bako

Academic Editor

PLOS ONE